# Relationships of Gut Microbiota Composition, Short-Chain Fatty Acids and Polyamines with the Pathological Response to Neoadjuvant Radiochemotherapy in Colorectal Cancer Patients

**DOI:** 10.3390/ijms22179549

**Published:** 2021-09-02

**Authors:** Lidia Sánchez-Alcoholado, Aurora Laborda-Illanes, Ana Otero, Rafael Ordóñez, Alicia González-González, Isaac Plaza-Andrades, Bruno Ramos-Molina, Jaime Gómez-Millán, María Isabel Queipo-Ortuño

**Affiliations:** 1Unidad de Gestión Clínica Intercentros de Oncología Médica, Hospitales Universitarios Regional y Virgen de la Victoria, Instituto de Investigación Biomédica de Málaga (IBIMA)-CIMES-UMA, 29010 Málaga, Spain; l.s.alcoholado@gmail.com (L.S.-A.); auroralabordaillanes@gmail.com (A.L.-I.); agonzalez.bq@gmail.com (A.G.-G.); isaacplazaandrade@gmail.com (I.P.-A.); 2Unidad de Gestión Clínica de Oncología Radioterápica, Hospital Universitario Virgen de la Victoria, 29010 Málaga, Spain; ana.otero.rom@gmail.com (A.O.); rafaelordm@gmail.com (R.O.); 3Grupo de Obesidad y Metabolismo, Instituto Murciano de Investigación Biosanitaria (IMIB-Arrixaca), 30120 Murcia, Spain; brunoramosmolina@gmail.com

**Keywords:** colorectal cancer, gut microbiota, SCFAs, gut permeability, radiochemotherapy, treatment outcome

## Abstract

Emerging evidence has suggested that dysbiosis of the gut microbiota may influence the drug efficacy of colorectal cancer (CRC) patients during cancer treatment by modulating drug metabolism and the host immune response. Moreover, gut microbiota can produce metabolites that may influence tumor proliferation and therapy responsiveness. In this study we have investigated the potential contribution of the gut microbiota and microbial-derived metabolites such as short chain fatty acids and polyamines to neoadjuvant radiochemotherapy (RCT) outcome in CRC patients. First, we established a profile for healthy gut microbiota by comparing the microbial diversity and composition between CRC patients and healthy controls. Second, our metagenomic analysis revealed that the gut microbiota composition of CRC patients was relatively stable over treatment time with neoadjuvant RCT. Nevertheless, treated patients who achieved clinical benefits from RTC (responders, R) had significantly higher microbial diversity and richness compared to non-responder patients (NR). Importantly, the fecal microbiota of the R was enriched in butyrate-producing bacteria and had significantly higher levels of acetic, butyric, isobutyric, and hexanoic acids than NR. In addition, NR patients exhibited higher serum levels of spermine and acetyl polyamines (oncometabolites related to CRC) as well as zonulin (gut permeability marker), and their gut microbiota was abundant in pro-inflammatory species. Finally, we identified a baseline consortium of five bacterial species that could potentially predict CRC treatment outcome. Overall, our results suggest that the gut microbiota may have an important role in the response to cancer therapies in CRC patients.

## 1. Introduction

Colorectal cancer (CRC) is the second most common malignant cancer in Western countries. The global burden of CRC is expected to substantially increase in the next two decades as a consequence of adopting Western lifestyles [1]. In recent years, several works have demonstrated that the gut microbiome could be a critical environmental factor that contributes to the tumorigenesis and progression of CRC, potentially by inducing pro-inflammatory responses, by producing microbial oncometabolites, and by interfering with the energy balance in cancer cells. Moreover, CRC is frequently associated with a dysbiosis in the microbial composition of the tumor and adjacent mucosa [2,3,4]. Several studies have suggested that the composition of the gut microbiota could affect the body’s response to a variety of cancer therapies, including chemotherapy, radiotherapy, and immunotherapy [5,6,7].

Preoperative radiochemotherapy (RCT) followed by surgery has become the standard treatment for patients with CRC [8,9]. Recent studies have suggested that the gut microbiota may influence drug response (efficacy and toxicity) in CRC patients through several mechanisms such as immunomodulation, reduced diversity, translocation, metabolism, and ecological variation [10]. Specific gut bacteria have been shown to affect cancer treatment by modulating drug metabolism and the host immune response [11,12]. Thus, several phyla are known to mediate drug metabolism via different reactions such as isoxazole scission, denitration, proteolytic degradation, acetylation/deacetylation, deconjugation, physical adherence to the drugs as well as by amine formation and/or hydrolysis [13]. Scott et al. described that the gut microbiota was able to influence the efficacy of one of the first-line treatments for CRC, such asfluoropyrimidines, through drug interconversion involving bacterial vitamin B6 and B9 and ribonucleotide metabolism [14]. In addition, the effect 5-fluorouracil treatment in CRC cells could be mediated by gut microbial metabolites [15]. Remarkably, *Fusobacterium nucleatum* is able to promote CRC resistance to chemotherapy by targeting both TLR4 and MYD88 innate immune signaling [16]. Furthermore, radiation may also lead to alterations in gut microbiota composition in animal models [17]. However, the clinical impact of radiotherapy on gut microbiota in cancer patients remains mostly unexplored although it has been proposed that the gut microbiota might play a role in the immunogenic effect of radiotherapy [18]. 

On the other hand, the gut microbiome produces bacteria-derived metabolites that could affect cancer proliferation and chemotherapy responsiveness. Thus, previous studies describe that SCFAs (such as butyric acid, isobutyric acid and acetic acid) inhibit the growth of cultured human colorectal cancer cells and that butyric acid is the strongest inhibitor [19]. Ross et al. reported an association between the levels of the short-chain fatty acids (SCFAs) propionate and butyrate in patients with early stage breast cancer with a pathological complete response (pCR) to neoadjuvant chemotherapy [20]. Coutzac et al. suggested that SCFA limits anti-CTLA-4 activity in patients with metastatic melanoma [21].

In addition, lower SCFA (especially butyrate) concentrations might induce a dysfunction in the gut epithelial barrier, thereby activating proinflammatory cytokines such as interleukin-6 (IL-6) and tumor necrosis factor-α (TNF-α), which damage epithelial cells and their junctions [22]. 

Other bacteria-derived metabolites, such as the polyamines (PAs) (spermine, spermidine and putrescine), have been involved in almost all the steps of colorectal tumorigenesis. PAs are molecules that are indispensable in normal cell growth and gene expression and are needed in cell proliferation, but their concentrations increase during the transition from a healthy cell to a tumor cell [23]. Recently, it was shown that the level of acetylated PAs is more specific for cancer. For example, N1, N12-diacetylspermine (DiAcSPM) was increased in CRC and in dysplastic colorectal lesions [24].

Therefore, taking all of the evidence together, we hypothesized a bidirectional interaction between the neoadjuvant RCT and the gut microbiome in CRC patients: RCT might induce alterations in the gut microbiome, and these alterations might, in turn, influence the effectiveness of RCT by directly interacting with the treatment and/or by stimulating the host’s immune response.

In this study, we aimed to identify the possible relationship between the gut microbiome, the fecal SCFAs levels, the serum levels of the polyamines and acetyl derivatives of polyamines, and the intestinal permeability to neoadjuvant RCT outcome in CRC patients. 

## 2. Results

### 2.1. Clinical Characteristics of the Patients and Healthy Controls

CRC patients and healthy controls had comparable eating habits to exclude the influence of dietary differences. CRC patients and healthy controls followed a Mediterranean diet consisting in a high consumption of olive oil, fruits, legumes, vegetables, nuts, whole grains, and fish and a low intake of red meat and dairy products. Adherence to the Mediterranean diet was assessed by using a validated 14-item food frequency questionnaire in all study patients. All CRC patients completed the neoadjuvant RCT and underwent surgical resection. There was no significant difference between CRC patients and healthy controls in terms of age, sex, BMI, and biochemical data (Table 1). A total of 28 of the 40 CRC patients (70%) had a good response to the neoadjuvant RCT (responders, R) (TGR 1–2), and 12 (30%) had a poor or non-response (non-responders, NR) (TGR 3–5) to therapy. Both R and NR patients were similar in terms of sex, age, BMI, and stage of the cancer, as shown in Table 1.

### 2.2. Differences in Taxonomic Composition and Diversity of Gut Microbiota between CRC Patients and Healthy Controls

The analysis of stool samples revealed 17,496,823 reads of the 16S rRNA gene (hypervariable V2–V9 regions), with an average of 105,632 (±10,825) reads for each sample in a range between 359 and 39,873. After trimming and filtering, 52,844 high-quality reads were selected. A total of 15,326 OTUs were obtained in the OTUs clustering process, and after the alignment of the OTU representative sequences, 2582 OTUs were identified to have a relative abundance >1% in at least four samples (97% similarity cut-off). For the taxonomic assignment of these OTUs, QIIME2 pipeline and Greengenes v13.8 were used, and the OTUs were binned into 7 phyla, 39 families, 45 genera, and 53 species.

We first compared the landscape of the gut microbiome in the stool samples of all CRC patients at baseline and in healthy controls in order to define a normal gut microbiota profile. As expected, we found significantly higher diversity and richness (defined by the Shannon and Chao1 indexes, respectively) in the fecal samples of healthy controls with respect to those of CRC patients (Shannon *p* = 0.026 and Chao1 *p* = 0.001) (Appendix AA,B). The beta diversity (Bray–Curtis dissimilarity) comparison of the baseline CRC patients and the healthy controls indicated that the two cohorts had significantly different genus compositions of intestinal bacteria (*p* = 0.0001, ANOSIM) (Appendix AC).

Furthermore, the analysis of the gut microbiota profiles between the CRC patients and the healthy controls at baseline revealed significant differences in the abundance at different taxonomic levels. At phylum level, the relative abundance of Fusobacteria (q < 0.001), Firmicutes (q < 0.001), Lentisphaerae (q = 0.007), and Proteobacteria (q = 0.003) were significantly increased in patients with CRC, while the relative abundance of Bacteroidetes (q < 0.001) and Actinobacteria (q = 0.034) were significantly decreased in CRC patients when compared to the controls (Figure 1A). 

At the genus level, the results indicated significant differences in the microbial composition of the dominant genera between the CRC patients at baseline and the healthy controls. Compared to the healthy controls, patients with CRC displayed an enrichment in the genera *Prevotella* (q < 0.001), *Oscillospira* (q < 0.001), *Fusobacterium* (q = 0.001), *Enterobacter* (q = 0.020), *Victivallis* (q = 0.012), *Escherichia* (q = 0.046), and *Desulfovibrio* (q < 0.001). Conversely, the abundance of *Bacteroides* (q = 0.003), *Roseburia* (q < 0.001), *Ruminococcus* (q = 0.006), *Faecalibacterium* (q = 0.01), *Bifidobacterium* (q = 0.023), and *Blautia* (q = 0.014) were enriched in the healthy controls compared to in the CRC patients (Figure 1B). 

At species level, while healthy subjects showed a significantly higher abundance of *Bifidobacterium bifidum* (q = 0.034) and *Faecalibacterium prausnitzii* (q = 0.040) with respect to the CRC patients, *Fusobacterium nucleatum* (q = 0.020), *Bacteroides fragilis* (q = 0.024), and *Escherichia coli* (q = 0.016) were significantly increased in the fecal samples of CRC patients in comparison to the controls.

### 2.3. Changes in Gut Microbiota Diversity and Composition in Response to Neoadjuvant RCT Treatment in CRC Patients

We compared the gut microbiota communities at baseline (T0) versus at post-treatment time points (T1, T2, and T3) to study the effect of neoadjuvant RCT on the gut microbial diversity and composition in CRC patients. The alpha diversity comparison showed no significant differences in the levels of richness (Chao 1) and diversity (Shannon) between the baseline and the different time points (Shannon *p* = 0.75 and Chao1 *p* = 0.61) (Figure 2A,B). Moreover, the PCoA plot based on the beta diversity (Bray–Curtis dissimilarity) revealed that the differences in the gut microbial community at T1, T2, and T3 compared to at baseline (T0) were not significant (*p* = 0.716, ANOSIM) (Figure 2C).

The main bacterial phyla (Firmicutes and Bacteroidetes) remained stable over time, while other, less abundant phyla, such as Fusobacteria and Proteobacteria, were significantly decreased at T3 compared to at T0 (q = 0.042 and q = 0.039, respectively) in the CRC patients. Although the bacterial family and genera proportions differed between the different time points, they were not significantly altered by the RCT treatment (Wilcoxon test *p* > 0.05), apart from the genera *Fusobacterium* (q = 0.015), *Escherichia* (q = 0.04) and *Klebsiella* (q = 0.035), which were significantly decreased after treatment, and the genus *Bifidobacterium* (q = 0.049), which was significantly increased at T3 compared to T0 (Figure 3).

### 2.4. Post-Treatment Microbiota Diversity and Composition Is Associated to Clinical Response to Neoadjuvant RCT in CRC Patients

To evaluate the relationship between the microbial community and the treatment outcome, we classified the patients based on their response to RCT into categories such as responders (R) and non-responders (NR). As shown in Table 1, no significant differences in terms of stage of cancer, sex, age, and BMI were observed between the study groups (R vs. NR).

An analysis of the alpha diversity at T3 revealed that the R group had higher diversity (Shannon index, q < 0.001; Simpson index, q = 0.039) and richness that the NR group (Chao1 index, q = 0.015) at genus level (Figure 4A,B). Furthermore, the ordination plot based on Bray–Curtis dissimilarities and the Jaccard index showed different intestinal microbial compositions at the genus level between both the R and the NR groups at T3 (Bray–Curtis index, q = 0.038; Jaccard index, q = 0.035; non-parametric ANOSIM test) (Figure 4C).

Next, we searched for differentially abundant taxa in the gut microbiome of R versus NR at T3. The analysis revealed that at the phylum level, there was a significant enrichment in the Actinobacteria (q = 0.0025) and Firmicutes (q = 0.0017) populations and a significant decrease in the Fusobacterias (q = 0.025) and Proteobacterias (q = 0.037) populations in the R group in comparison to the NR group (Figure 5A,B). At the family level, a significantly higher abundance of Ruminococcaceae (q = 0.004) and Bifidobacteriaceae (q = 0.03) accompanied with a significantly lower abundance of Prevotellaceae (q = 0.045), Enterobactericeae (q = 0.027), and Fusobacteriaceae (q = 0.014) were shown in the R group compared to in the NR group (Figure 5A,C).

In addition, at the genera level, we identified a significant increase in *Ruminococcus* (q = 0.035), *Bilophila* (q = 0.008), *Collinsiella* (q = 0.015), *Bifidobacterium* (q = 0.024), *Roseburia* (q = 0.032), and *Faecalibacterium* (q = 0.041) in R patients with respect to the NR, while a significant increase in *Prevotella* (q = 0.05), *Fusobacterium* (q = 0.045), *Escherichia* (q = 0.037), *Bacteroides* (q = 0.027), and *Klebsiella* (q = 0.035) were observed in the NR patients compared to the R group (Figure 6A,B). Finally, at the species level, we found a significant overabundance of *Prevotella copri* (q < 0.001), *Escherichia coli* (q = 0.029), *Fusobacterium nucleatum* (q = 0.015), and *Bacteroides fragilis* (q = 0.029) in the NR group, while the R group displayed a significantly higher abundance of *Bifidobacterium bifidum* (q = 0.043), *Ruminococcus albus* (q = 0.019), *Collinsella aerofaciens* (q = 0.018), and *Faecalibacterium prausnitzii* (q = 0.027).

### 2.5. Baseline Microbiota Composition Could Predict Response to RCT Treatment in CRC Patients 

After describing the significant differences in the intestinal microbial composition between the R and NR after RCT treatment, we next assessed the predictive power of the gut microbiome related to neoadjuvant RCT response. We used random forest (RF) to build a predictive model based on the overall gut microbiota profile using the species-level abundance data as input. After RF analysis with 500 bootstrap samples, we found that the overall gut microbiota composition data had a significant accuracy of 80% and an area under the curve (AUC) of 0.71. The main species accounting for this stratification were *Ruminococcus albus*, *Bifidobacterium bifidum*, *Faecalibacterium prausnitzii*, *Fusobacterium nucleatum*, and *Bacteroides fragilis*, and when the proportions of these bacterial species were only used for testing the accuracy of the RF classifier, this increased to 96% (AUC = 0.92). Thus, the response to RTC or the lack of it were identified with an accuracy of 94% (AUC = 0.95) and of 91% (AUC = 0.92), respectively (Figure 7A). The validation cohort consisted of 84 CRC patients under neoadjuvant RCT (45 R patients and 39 NR patients) (data collected from the Genome Sequence Archive in National Genomics Data Center, accession number CRA002850). After RF analysis in this validation cohort, an accuracy of 92.0% (AUC = 0.93) and 90.0% (AUC = 0.91) were obtained for the response to RTC or the lack of it, respectively (Figure 7B). Among the five species variables, *Fusobacterium nucleatum*, and *Bacteroides fragilis* were biomarkers of R patients, and *Ruminococcus albus*, *Bifidobacterium bifidum*, and *Faecalibacterium prausnitzii* were biomarkers of NR patients.

### 2.6. Differences in the Gut Microbiota Functions between Responder and Non-Responder

KEGG pathway enrichment analysis of the metagenomic data showed that genes for energy metabolism such as methane metabolism (q < 0.004), carbohydrate metabolism, such as the pentose phosphate pathway (q = 0.0022), pyruvate metabolism (q-value < 0.001), starch and sucrose metabolism (q = 0.008), galactose metabolism (q = 0.007), butanoate metabolism (q = 0.005), and glycolysis-gluconeogenesis (q = 0.0028); for xenobiotic biodegradation and metabolism pathways, including benzoate degradation (q = 0.038) and nitrotoluene degradation (q = 0.005); and membrane transport, such as ABC transporters (q = 0.012) and transporters (q = 0.012), were significantly depleted in NR compared to R patients. 

Nevertheless, compared to the R patients, in the NR patients, there was a significant over-representation of genes for lipid metabolism, such as for araquidonic acid metabolism (q = 0.006); amino acid metabolism pathways, such as for arginine and proline metabolism (q = 0.029); for glycine, serine, and threonine metabolism (q = 0.001); in genes for the metabolism of other amino acids such as glutathione metabolism (q = 0.003); for the metabolism of cofactors and vitamins such as riboflavin metabolism (q = 0.003), ubiquinone, and other terpenoid metabolism (q < 0.001); folate biosynthesis (q = 0.014), glycan biosynthesis, and metabolism, such as lipopolysaccharide biosynthesis (q = 0.007); lipopolysaccharide biosynthesis proteins (q = 0.001); cellular processes and signaling that contain cell motility and secretion (q = 0.0018); oxidative phosphorylation (q < 0.001); and for pathways in cancer (q < 0.001) (Figure 8).

### 2.7. Changes in the Serum Level of Polyamines and Zonulin and Fecal Levels of SCFAs after RCT Treatment in CRC Patients

Significant differences in the serum levels of several polyamines and acetyl derivatives of polyamines were found in the R and NR patients at post-treatment point (T3). Then, in the NR patients, we found a significant increase in the levels of spermine, N1-acetyl spermine (N1-AcSP), N1, N12-diacetylspermine (N1, N12-DiAcSP), N1-acetylspermidine (N1-AcSPD), N1, N8- diacetylspermidine (N1, N8-DiAcSPD), and N1-acetylputrescine (N1-AcPUT) compared to those in the R patients. On the other hand, within-group, there were also significant changes in the levels of N1-AcSPD and spermine in both the R and NR patients and in the serum levels of N8-AcSPD only in the NR group (Table 2).

SCFAs are bacterial-derived metabolites with important physiological functions in the host and that have anti-cancer properties. Analysis of the fecal levels of SCFAs revealed significant differences in the concentrations of acetic, butyric, isobutyric, valeric, isovaleric, and hexanoic acid between the R and NR study groups at post-treatment time point T3. Moreover, we found several significant differences in the within-group comparison of the fecal concentrations of acetic and butyric acid, which significantly increased after RCT treatment in the R group. On the other hand, serum zonulin levels (a circulating marker of gut permeability) were significantly increased in the NR group (but not in R group) after RCT treatment (Table 3). 

In addition, pairwise comparisons using Spearman rank correlation analysis were then performed between bacterial species enriched in the gut microbiome of both the R and NR patients and the fecal SCFAs and serum polyamines and zonulin levels. Interestingly, we found a statistically significant positive correlation between the fecal levels of butyrate and the abundance of the *Faecalibacterium prausnitzii* (r = 0.816 *p* < 0.001) and *Ruminoccocus albus* (r = 0.924 *p* = 0.008) in the R group and between the concentration of propionic acid and *Bacteroides fragilis* in the NR group. In addition, negative associations *of Faecalibacterium prausnitzii* with the serum levels of spermine (r = −0.619 *p* = 0.018) and N1,N12-DiAcSP (r = −0.793 *p* = 0.01) in the R patients were described, while there was a positive association between the abundance of Bacteroides fragilis and *Fusobacterium nucleatum* with the levels of N1,N12-DiAcSP (r=0.436 *p* = 0.043; r = 0.637 *p* = 0.001, respectively) and N8-AcSPD (r = 0.547 *p* = 0.014; r = 0.752 *p* < 0.001) in the NR patients. Finally, *Prevotella copri* was positively associated with the serum zonulin levels in NR patients.

## 3. Discussion

In this study, we have demonstrated the existence of a significant association between the gut microbiota and the anti-cancer response of CRC patients treated with neoadjuvant RCT. Moreover, we have found that some microbial-derived metabolites such as SCFAs could be at least partially responsible for the response to RCT in these CRC patients. Finally, we have identified a baseline consortium of CRC-enriched bacterial species that may potentially serve as diagnostic bacterial markers of a good or bad response to neoadjuvant RCT. Where *Ruminococcus albus, Bifidobacterium bifidum,* and *Faecalibacterium prausnitzii*, were overrepresented in R patients and chosen as discriminatory variables in our response-prediction RF model, *Fusobacterium nucleatum* and *Bacteroides fragilis* were overrepresented in the NR patients.

The loss of microbial diversity has been associated with chronic health conditions [25,26,27] and cancer [27,28] as well as with poor outcomes to certain forms of cancer therapy [29,30,31]. Accordingly, recent works have also reported that patients with CRC display a lower bacterial diversity and richness in fecal samples and the intestinal mucosa compared to healthy individuals [32,33]. In this study, we found that compared to healthy controls, the CRC microbiota exhibited a state of dysbiosis with a reduced overall bacterial richness and diversity. In addition, the analysis of the Bray–Curtis PCoA plot for beta diversity revealed that the CRC patients were clustered separately to the healthy controls, suggesting important CRC-mediated microbial changes. 

Related to gut microbiota composition, several microbes have been found to be differentially represented in fecal samples between both study groups. Thus, the gut microbiota in the CRC patients was enriched with pro-inflammatory opportunistic pathogens and was depleted in butyrate-producing bacteria, which have been shown to be essential for the preservation of intestinal homeostasis [34]. In particular, we have shown that some bacteria such as *Fusobacterium nucleatum*, *Escherichia coli*, and *Bacteroides fragilis* had high prevalence in CRC patients in comparison to the healthy controls, whereas genera such as *Roseburia*, *Faecalibacterium*, and *Bifidobacterium* were depleted, demonstrating that microbial dysbiosis was already present in CRC at the time of diagnosis. 

On the other hand, we observed that gut microbiota composition was relatively stable over treatment time following RCT treatment, with the exception of a significant decrease in the abundance of *Fusobacterium*, *Escherichia*, and *Klebsiella* and a significant increase in *Bifidobacterium* (probiotic bacteria) at post-treatment time compared to at baseline, showing the beneficial effect of RCT on the gut microbiome of CRC patients. *Klebsiella* and *Fusobacterium* are pathogens normally found in the human intestine that cause diarrhea and bloodstream infections and that considerably increase the rates of treatment failure and death [35]. 

After treatment, the CRC patients were classified as responders (N) versus non-responders (NR), based on their good or poor response to the RCT. Interestingly, we found significant differences in the alpha diversity at the genus level, with an increase in the diversity (Shannon) and richness (Chao 1) in the R patients compared to in the NR patients. Similarly, there was a statistically significant difference in B-diversity (Bray–Curtis dissimilarities and Jaccard index), finding a notable clustering effect by response status in the gut microbiome of these patients, indicating a difference in the bacterial community composition between the R and NR patients. 

At the taxa levels, we found a significant enrichment in probiotic and butyrate producer-bacteria such as *Bifidobacterium bifidum*, *Ruminoccous albus*, *Roseburia*, and *Faecalibacterium prausnitzii* in the R patients, while the NR patients showed an enrichment in unfavorable microbial taxa such as *Fusobacterium nucleatum*, *Bacteroides fragilis*, *Escherichia coli*, *Prevotella copri,* and *Klebsiella*. Several studies have shown that butyrate-producing bacteria are negatively related to irritable bowel disease and colorectal cancer [36,37]. 

Additionally, both *Fusobacterium* and *Prevotella* have been related to recurrent CRC after chemotherapy. Given that *Fusobacterium nucleatum* has been previously correlated with chemoresistance [17], our results may suggest that the higher load of *Fusobacterium nucleatum* present in NR patients could be a potential promoter of CRC chemoresistance and therefore of a poor response to CRC treatment. Similarly, the enterotoxigenic *Bacteroides fragilis*, which was also enriched in the NR patients, is a significant source of chronic inflammation, and it has previously been associated with the development and aggressiveness of colorectal cancer and poor patient outcome [6,38]. These data also suggest that the gut microbiota composition of the R patients shifted towards a microbial profile that has great similarity to the gut microbiota of the healthy controls.

Next, we sought to gain insight into the mechanism through which the gut microbiome may influence response to RCT. Regarding the metabolic function of gut microbiota, in the current study, Picrust analysis showed significant differences between the R and NR patients. In the NR patients, we have found an increase in the abundance of genes for lipopolysaccharide biosynthesis as well as for araquidonic acid metabolism, for glutathione metabolism, and for the amino acid metabolism pathways (such as arginine and proline metabolism) compared to in the R patients. The significant increase of genes for lipopolysaccharide biosynthesis could be related to the significant increase in the abundance of Gram-negative bacteria such as *Escherichia coli* in the NR patients; these bacteria contain specific enzymes that produce LPS, which can induce Toll-like receptor 4 signaling and can promote cell survival and proliferation in CRC patients [39]. Similarly, the arachidonic acid pathway is important in the development and progression of numerous malignant diseases, including CRC, due to the fact that araquidonic acid stimulates key downstream signaling cascades that regulate cell proliferation, apoptosis, angiogenesis, inflammation, and immune surveillance [40,41]. With respect to the increase in the genes for glutathione metabolism in NR patients, some studies have described that the elevated levels of glutathione in tumor cells are able to protect such cells in bone marrow, breast, colon, larynx, and lung cancers by conferring resistance to several chemotherapeutic drugs [42,43]. Other bacterial functions involving the metabolism of cofactors and vitamins and the energy production pathways such as oxidative phosphorylation were also increased in NR patients. These pathways may serve as alternative bioenergetic sources for metabolically stressed cancer cells [44]. 

Remarkably, a recent metagenomic analysis reported that the CRC-associated microbiome showed an association with the conversion of amino acids into polyamines (e.g., the biosynthesis of putrescine from the amino acids L-arginine and L-ornithine), indicating that these metabolites could be particularly important in CRC development and progression [45]. In our study, significant differences in the serum levels of several polyamines and acetyl derivatives of polyamines were found between R and NR patients at post-treatment point. Moreover, we observed that the abundance of N1,N12-DiAcSP and N8-AcSPD were positively associated with the increased abundance of *Bacteroides fragilis* and *Fusobacterium nucleatum* in NR patients. 

In fact, Bacteroides spp. and Fusobacterium spp. can synthesize putrescine and spermidine in vitro and in vivo [46]. Goodwin et al. demonstrated that the purified *Bacteroides fragilis* toxin (BFT) up-regulates spermine oxidase in HT29/c1 and T84 colonic epithelial cells, producing the spermine oxidase-dependent generation of ROS and the induction of a marker of DNA damage such as γ-H2A.x. [47]. In another study, Johnson et al. found that antibiotic treatment led to a lower tissue concentration of N1, N12-diacylspermine and that a disturbed bacterial biofilm was observed in resected CRC tissues compare to CRC tissues with negative bacterial biofilm, suggesting the implication of gut microbes in the increase of host generated N1, N12-diacetylspermine [48]. Moreover, the activation of the amino acid metabolic pathways by the intestinal microbiota of the NR patients could contribute to the increase in polyamines, which are actively assimilated by the cells of the intestinal epithelium and induce rapid cell proliferation, favoring the tumorigenesis [49,50].

On the other hand, several works performed in both cellular and animal models have demonstrated that CRC is linked to alterations in the metabolism of SCFAs, which have been shown to exhibit potential anti-carcinogenic effects [51,52]. Here, we have found that R patients displayed a significant over-representation of genes involved in butanoate metabolism and a significant increase in the fecal abundance of several SCFAs such as acetic and butyric acid after RCT treatment. Moreover, there was a positive correlation between the fecal levels of butyrate and the abundance of *Faecalibacterium prausnitzii* and *Ruminoccocus albus* in these patients. *Faecalibacterium praustnitzi* is considered important in health promotion, as it is able to produce butyrate from dietary fibre and possesses anti-inflammatory properties [53]. A decrease in *Faecalibacterium prausnitzii* and butyrate levels defines microbiota dysbiosis in patients suffering inflammatory bowel disease [54]. In addition, *Faecalibacterium* is able to use the acetate produced by *Bifidobacterium* (also increased in N patients) with the subsequent modulation of the intestinal mucus barrier by the modification of goblet cells and mucin glycosylation [55]. Butyrate is required for colonic epithelium repair and the production of Treg cells, which regulate the local immune response and suppress colonic inflammation and carcinogenesis [56]. Moreover, butyrate has been described to be able to induce the production of IL-18 by the intestinal epithelial cells through the activation of the GPR109a receptor, which stimulates mucosal tissue repair via the regulation of the production and availability of IL-22 [57]. The absence of IL-18 has been associated with gut microbiota dysbiosis, a dysregulation of the homeostatic and mucosal repair and alteration of the inflammatory response, producing an increased susceptibility to carcinogenesis [58]. In addition, after RCT treatment, we found a significant decrease in the fecal levels of acetic, butyric, isobutyric, and hexanoic acid in the NR study group compared to in R patients, indicating the exhaustion of butyric acid-producing microbiota in their colon. In a previous study, hexanoic acid was shown to reduce the colonization and dysbiotic expansion of potentially pathogenic bacteria in the gut [59]. 

Finally, we found that plasma zonulin levels were significantly increased in the NR patients compared to in the R patients. A higher zonulin level was correlated with the relaive abundance of *Prevotella copri* in the R patients. Zonulin is a protein synthesized in intestinal and liver cells that reversibly modulates the intestinal permeability of the intestinal epithelial barrier by modulating intercellular tight junctions [60]. Wright et al. found that *Prevotella* contains key enzymes implicated in mucin degradation, which are able to disrupt the colonic mucus barrier. A disrupted mucosal barrier may result in increased intestinal permeability, which allows the diffusion of antigens, toxins, and pathogens from the luminal environment into the mucosal tissues and circulatory system [55]. As a consequence, an inflammatory response can be triggered that induces cancer initiation, progression, and response to anticancer treatment [61]. Then, the significant increase in *Prevotella* abundance found in our study could be associated in party with the poor or non-response to RCT in NR patients.

This study has some limitations, such as the relatively small sample size, which could reduce the power of the study. However, despite the relatively small size of our study, statistically significant differences were observed, suggesting that the results presented herein provide solid evidence on the potential contribution of the gut microbiome to RCT outcomes in CRC patients. Moreover, our study also has several strengths, such as the careful design, the well-matched cohorts of CRC patients and controls, a complete definition of the inclusion and exclusion criteria, and the consideration of lifestyle-associated confounding factors that may affect the gut microbiota composition, such as dietary pattern. 

## 4. Materials and Methods

### 4.1. Study Patients

A total of forty patients aged 35–75-years-old who were newly diagnosed with CRC in stages II–III (T2–T4 and/or N1–N2) from the Radiotherapy Oncology Service at the Virgen de la Victoria Hospital and with no metastatic lesions detected on imaging were enrolled in the study and were followed-up with for at least 1 year. All of the CRC patients received only neoadjuvant treatment for 5 weeks with pelvic radiation therapy (50 Gy in fractions of 2 Gy/session) and oral capecitabine (825 mg/m^2^/12 h) during radiotherapy treatment. Patients with a history of colorectal cancer or bowel resection, type 2 diabetes, chronic inflammatory bowel disease, severe active infection, or hereditary colorectal cancer syndromes were excluded from the study. Patients who received pelvic cancer radiation therapy or anti-tumor treatment in the previous 2 years, who used antibiotics or immunosuppressants in the previous 2 months, or who regularly used non-steroidal anti-inflammatory drugs, statins, or probiotics before the study were also excluded. A pathologist examined surgical specimens and tumor response after neoadjuvant RCT was determined in surgical specimens according to the tumor regression grades (TRG) system described by Mandard et al. [62]. We divided the CRC patients into TRG1–2 (patients with good response or responders (R)) and TRG 3–5 (patients with poor or non-response (NR)). Blood and fecal samples were collected at baseline (T0), 2 and 4 weeks after starting RCT (T1 and T2, respectively), and 7 weeks after finishing treatment (T3).

In the study, we also included fecal samples from 20 healthy patients that were matched with the CRC patients according to sex, age, and BMI. The healthy controls did not have gastrointestinal tract disorders or other complications and were not administered antibiotics or probiotics during the 2 months prior to sample collection.

The study protocol was approved by the Medical Ethics Committee at the Virgen de la Victoria University Hospital and was conducted in accordance with the Declaration of Helsinki. Written informed consent was provided by all study participants.

### 4.2. Laboratory Measurements

Fasting venous blood samples were collected, and serum was separated in aliquots and was immediately frozen at −80 °C. Serum levels of glucose, total cholesterol, triglycerides, HDL-cholesterol, and LDL-cholesterol were measured in duplicate using a Dimension autoanalyzer (Dade Behring Inc., Deerfield, IL, USA) using enzymatic methods (Randox Laboratories Ltd. Ardmore, UK). 

### 4.3. DNA Extraction and Gut Microbiota Sequencing

The frozen fecal samples were thawed at 4 °C to avoid dramatic temperature changes that might affect bacterial DNA integrity. Afterwards, the fecal samples were manually homogenized for 30 s with a sterile plastic scoop, and aliquots of 200 mg were used for DNA extraction using the QIAamp DNA Stool Mini kit following the manufacturer’s instructions (Qiagen, Hilden, Germany). DNA concentration (A260) and purity (A260/A280 ratio) were estimated with a Nanodrop spectrophotometer (Nanodrop Technologies, Wilmington, DE, USA).

DNA was amplified using the Ion 16S Metagenomics kit (Thermo Fisher Scientific, Madrid, Spain), which contains a primer pool to amplify multiple variable regions (V2, 3, 4, 6–7, 8 and 9) of the 16S rRNA gene. The Ion PlusTM Fragment Library Kit (Thermo Fisher Scientific, Madrid, Spain) was used to ligate the barcoded adapters to the generated amplicons and to create the barcoded libraries, which were pooled and templated on the automated Ion Chef system (Thermo Fisher Scientific, Madrid, Spain). The sequencing was done on an Ion S5 platform (Thermo Fisher Scientific, Madrid, Spain).

### 4.4. Bioinformatics Analysis

Analysis of 16S rRNA amplicons was performed using QIIME (2-2019.4 version). The q-dada2 plugin with the DADA2 pipeline was used for the quality filtering and the denoised, dereplicated, and chimera filtering of the raw sequence data. The sequence variants obtained through the DADA2 pipeline were merged into a single feature table using the q2-feature-table plugin. Using the q2-vsearch plugin with 97% sequence similarity, all amplicon sequence variants from the merged feature table were clustered into OTU’s using the Open Reference Clustering method against Greengenes version 13_8 with 97% similarity from the OTU reference sequences. The OTUs were aligned with MAFFT (via q2-alignment) and were used to construct a phylogeny with fasttree2 (via q2-phylogeny). The q2-feature-classifier classify-sklearn naive Bayes taxonomy classifier was used to assign taxonomy to the OTUs. Alpha diversity metrics (Shannon and Chao1), beta diversity metrics (Bray–Curtis dissimilarity), and principal coordinate analysis (PCoA) were estimated using a q2-diversity plugin after the samples were rarefied to 994 sequences per sample. Alpha diversity significance was estimated with Kruskal–Wallis test, and beta diversity significance was estimated using the non-parametric ANOSIM test.

### 4.5. Analysis of Short-Chain Fatty Acids (SCFAs) in Fecal Samples by Gas Chromatography (GC) Coupled with a Flame-Ionization Detector 

The fecal concentrations of SCFAs were measured by GC coupled with a flame-ionization detector as previously described [63,64,65,66] in the Servicios de Apoyo a la Investigación de la Universidad de Extremadura (SAIUEx). Briefly, 20 mg of the fecal samples were homogenized manually using a spatula in 200 μL of distilled water. Subsequently, 100 μL of homogenized fecal samples were mixed with 40 mg of sodium chloride, 20 mg of citric acid, 40 μL of 0.1 M hydrochloric acid, and 200 μL of butanol: tetrahydrofuran: acetonitrile (50:30:20). The samples were then vigorously vortexed for 3 min and were centrifuged at 14,870× *g* at room temperature for 10 min. The supernatant was transferred to a new plastic tube, and 200 μL of a benzyl alcohol–pyridine mixture (3:2) and 100 μL DMSO were added, and the mixture was vortexed for 5 s. Then, 100 μL of benzyl chloroformate was added carefully. To release the gases generated by the reaction, the tube lid was kept open for 1 min. The tube was then closed, and the mixture was vortexed. After derivatization, 200 μL hexane was added to the reaction mixture, and the sample was vortexed for 5 min followed by a centrifugation step at 21,000× *g* for 2 min. Subsequently, 100 μL of derivative extract (upper hexane layer) was transferred to a glass insert, and 5 μL were injected into the gas chromatograph and were further analyzed using an Agilent 6850 gas chromatograph coupled with a split/spitless injector and a flame-ionization detector (FID) (Agilent Technologies, Santa Clara, CA, USA). 

The temperature of the injector and detector was adjusted to 250 °C, and the samples (5 μL) were injected in a split ratio of 25:1 using a fused-silica capillary DB-23 column Agilent (60 m × 0.25 mm (internal diameter) coated with a 0.15 μm thick layer of 80.2% 1-methylnaphatalene. Nitrogen was used as the carrier gas at 1 mL/min (hold 4 min), reduced to 0.8 mL/min (hold 1 min) and then 0.6 mL/min (hold 1 min), and finally increased to 1 mL/min. The temperature of the FID detector was adjusted and maintained at 260 °C, and the flow rates of H_2_, the air, and the make-up gas N_2_ were adjusted to 30 mL/min, 350 mL/min, and 25 mL/min, respectively. The initial oven temperature was 100 °C (hold 2 min), which was increased to 200 °C at a rate of 15 °C/min, and was finally maintained at 200 °C for 5 min. The identity of the SCFAs detected in the fecal samples was confirmed through the comparison of their retention times and their mass spectra with those of the analytical SCFA standards (Sigma–Aldrich, Madrid, Spain). The standard calibration curves for SCFAs (acetic acid, propionic acid, butyric acid, isobutyric acid, valeric acid, isovaleric acid, 4- methylvaleric acid, hexanoic acid, and heptanoic acid) were prepared in triplicate, with a concentration range of 15–1,000 μg/mL.

### 4.6. Analysis of Serum Polyamine Levels by Ultra-High Performance Liquid Chromatography Tandem Mass Spectrometry (UHPLC-MS/MS) 

For the analysis of the polyamine concentrations, serum samples were processed as previously described [67]. Briefly, 50 μL of serum (aliquoted in 1.5 mL Eppendorf LoBind tube) were mixed with 5 μL of internal standard and 167 μL of methanol. The mixture was vortexed for 1 min, and 334 μL of chloroform was added, vortexed for 1 min, and centrifuged for 10 min at 15,000 rpm and 4 °C. After centrifugation, the upper layer was collected and was transferred to a new tube, where 100 μL of carbonate–bicarbonate buffer (pH 9) and 50 μL of dansyl chloride (10 mg/mL in acetone) were added to derivatize the sample. The mixture was vortexed and was placed in the dark for 1 h at room temperature. A total of two extractions of the compounds were conducted with 250 μL of ethyl acetate, between which 2.5 μL of trifluoroacetic acid were added. A SpeedVac at 45 °C was used to evaporat the combined organic phases, which were stored at −20 °C until analysis. An amount of 50 μL of ammonium acetate and 0.2 M acetonitrile (30:70) was used to reconstitute the samples.

Chromatography of the samples was completed with Agilent UHPLC 1290 series binary pump equipment (Agilent Technologies, Santa Clara, CA, USA), and the separation was performed on a Kinetex EVO C18 column (2.6 μm particle size, 2.1 mm internal diameter × 150 mm length) (Phenomenex, Torrance, CA, USA) held at 25 °C. A gradient was established between the water acidified with 0.1% formic acid (A), and acetonitrile acidified with 0.1% formic acid (B) at a flow rate of 400 μL/min was used as a mobile phase for elution. The injected amount was 2.5 μL.

MS/MS analysis was conducted in an Agilent QqQ 6490 Series mass spectrometer operating in AJS + ESI. The optimization of the ionization source parameters was performed as follows: nebulizer gas (nitrogen) with a pressure of 15 psi, a gas flow of 15 L/min at 200 °C, a sheath gas flow of 11 L/min at 350 °C, a capillary voltage of 2.5 kV, and a nozzle voltage of 1000 V in a MassHunter Optimizer (Agilent Technologies, version 6.0)

An Agilent UHPLC 1290 Infinity II Series coupled to an Agilent QqQ/MS 6490 Series (Agilent Technologies, Sta. Clara, CA, USA) was used for LC-MS/MS analysis, while chromatographic separation was performed using a Kinetex EVO C18 analytical column (2.6 μm; 2.1 mm × 150 mm) (Phenomenex, Torrance, CA, USA)

Quantification was completed with the commercial standards ornithine, spermine, arginine, spermidine, putrescine, N1-acetylspermidine, N8-acetylspermidine, N1-acetylspermine, N1-acetylputrescine, N1,N8-diacetylspermidine, and N1,N12-diacetylspermine (Toronto Research Chemicals, North York, ON, Canada). The internal standards of the amino acids were arginine (13C6, 15N4) and lysine (13C6, 15N2) (Cambridge Isotope Laboratories), and for the polyamines, the internal stanfdards comprised putrescine-d8, spermidine-d6, spermine-d20, and N8-acetylspermidine-d3 (Toronto Research Chemicals).

### 4.7. Intestinal Permeability Analysis

Plasma levels of zonulin were measured in duplicate using an ELISA commercial kit (Immunodiagnostik AG, Bensheim, Germany). Mean values were used for data analysis. Intra- and inter-assay coefficients of variation were between 3–10%, and the detection limit was 0.22 ng/mL.

### 4.8. Statistical Analysis

The Kruskal–Wallis rank-sum test was performed to compare the bacterial abundance between the study groups, and the false discovery rate (FDR) using the Benjamini–Hochberg method was applied to correct the significant p-values (q < 0.05). The Kruskal–Wallis rank-sum test and subsequent post hoc Bonferroni were used to analyze differences in the clinical and biochemical variables between three study groups, whereas differences between the two groups were analyzed using the Mann–Whitney U test. Inter-group comparison among post-treatment changes in fecal SCFAs and plasma zonulin levels were performed using a covariance model (ANCOVA) adjusted for baseline. A Wilcoxon signed-rank test was used to calculate differences in fecal SCFAs and plasma zonulin between baseline and the post-treatment timepoint T3. The Spearman correlation coefficients were calculated to estimate the correlations between the bacterial taxa and microbial derived-metabolites (SCFAs and polyamines) and the permeability. Statistical analyses were conducted with the statistical software package SPSS version 26.0 (SPSS Inc., Chicago, IL, USA). Random forests (RF) were used to predict baseline bacteria (species-level relative abundance data) related to the neoadjuvant RCT response using the default parameters of the R implementation of the algorithm (R package “randomForest”), and bootstrapping (n = 500) was used to assess the classification accuracy. P values below 0.05 were considered statistically significant.

## 5. Conclusions

In this study, we have demonstrated that the gut microbiota in CRC patients differs in intestinal microbiota composition in comparison with healthy controls. In CRC patients, the gut microbiota is characterized by a significantly lower bacterial diversity and richness, a significant increase in proinflammatory opportunistic pathogens, and a decrease in the relative abundance of beneficial or commensal butyrate-producing bacteria.

In addition, neoadjuvant RCT treatment did not induce significant changes in gut microbiota diversity and composition, with the exception of a significant decrease in *Fusobacterium*, *Escherichia,* and *Klebsiella* and a significant increase in *Bifidobacterium* at post-treatment time compared to baseline. Nevertheless, after the classification of CRC patients in the R and NR groups to the neoadjuvant RCT, we observed a significant increase in the diversity and richness in R patients compared to in the NR patients. Additionally, a compositional change was shown between both study patient groups, with a significant enrichment of probiotic and butyrate-producing bacteria in the R patients, accompanied by an enrichment in unfavorable pro-inflammatory bacteria in the NR patients. Moreover, the NR patients had significantly higher levels of spermine and some acetyl derivatives of polyamines and serum zonulin and significantly lower levels of fecal of acetic, butyric, isobutyric, and hexanoic acids than the R patients. These microbial-derived metabolites are important factors that connect the intestinal microbiota to CRC and could be responsible for RCT efficiency. Moreover, in the NR patients, the PICRUSt analysis found an over-representation of genes involved in lipopolysaccharide biosynthesis as well as in araquidonic acid and glutathione metabolism, genes from pathways associated with bacterial pathogenesis, inflammation, cell survival, proliferation, and therapy response. 

In addition, we also identified a baseline consortium of CRC-enriched bacterial species (*Ruminococcus albus, Bifidobacterium bifidum, Faecalibacterium prausnitzii, Fusobacterium nucleatum*, and *Bacteroides fragilis*) that potentially could predict cancer treatment outcome, suggesting that the intestinal composition in CRC patients is important in predicting the response of the gut microbiome to neoadjuvant RCT. Altogether, our results suggest that a healthy gut microbiome could be indispensable for an optimum therapeutic response and that dysbiotic microbiota could be the underlying reason for variable responses to similar therapeutic strategies in different patients.

## Figures and Tables

**Figure 1 ijms-22-09549-f001:**
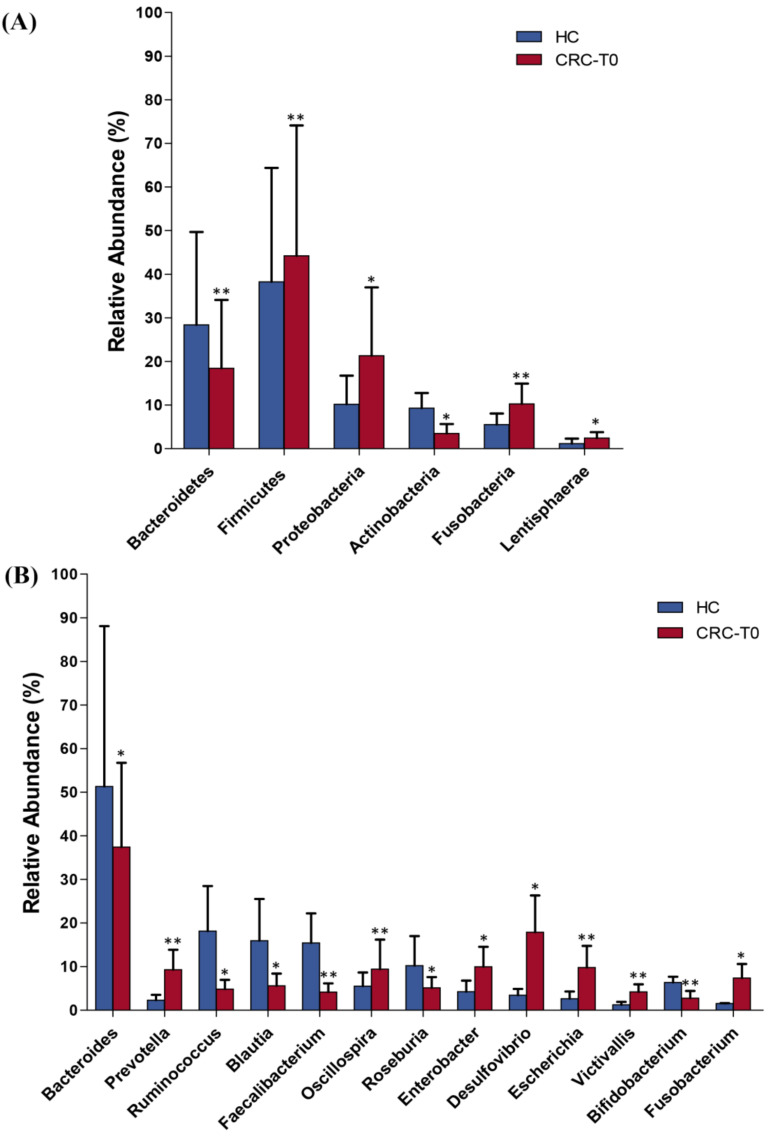
Relative abundance at phylum (**A**) and genera (**B**) levels of differentially abundant bacteria in the stool samples of CRC patients at baseline (CRC-T0) and healthy controls (HC). * *p* < 0.05, ** *p* < 0.001.

**Figure 2 ijms-22-09549-f002:**
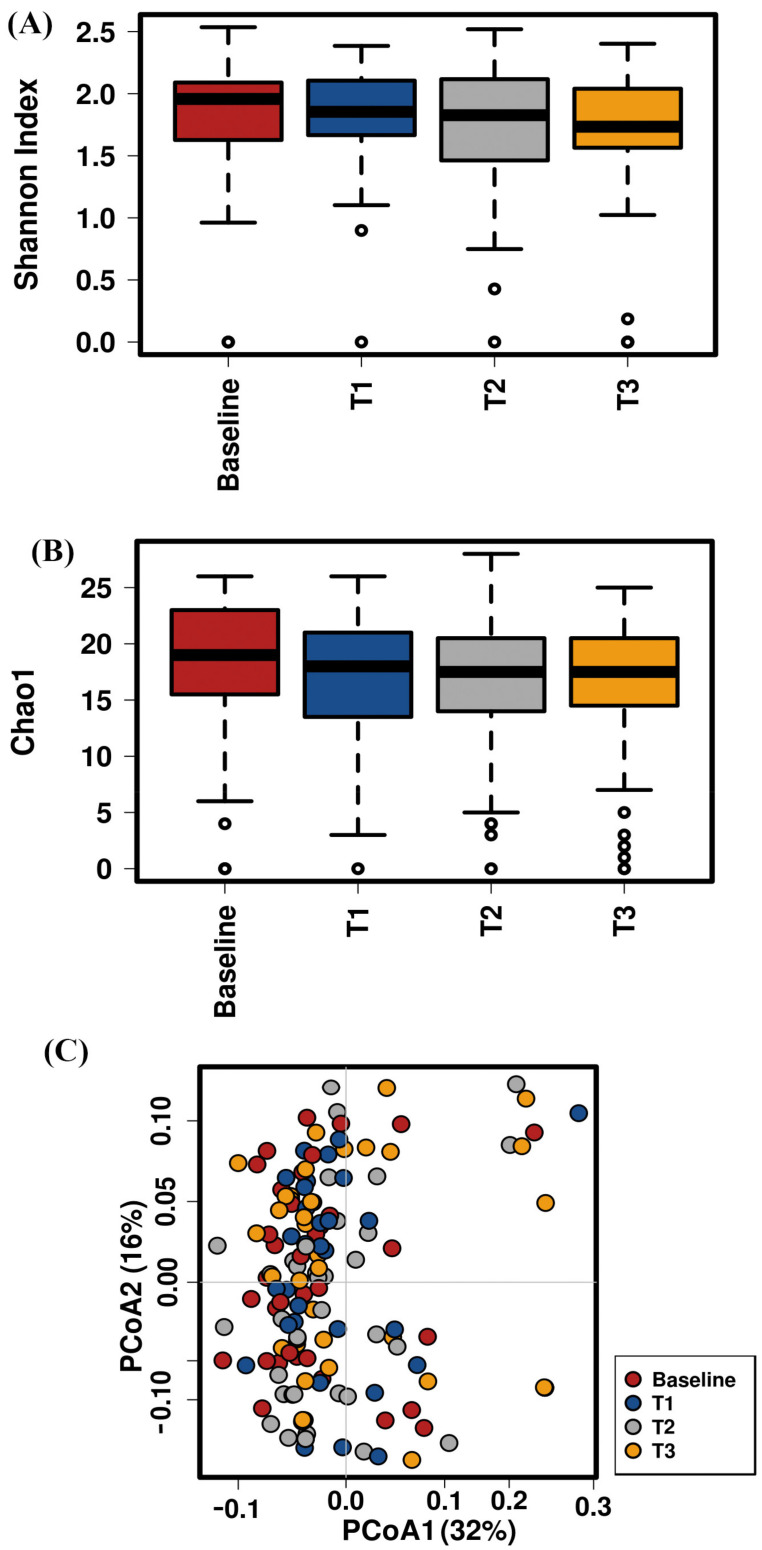
Gut microbiota diversity and richness at baseline and during RTC treatment and post-treatment points in CRC patients. (**A**) Shannon index (*p* = 0.75); (**B**) Chao1 index (*p* = 0.61); (**C**) principal component analysis representation based on Bray–Curtis distance matrix of patient distribution based on bacterial genera composition at baseline and during RTC treatment and at post-treatment points (*p* = 0.716). The first two coordinates are plotted with the percentage of variability, which is explained and indicated on the axis.

**Figure 3 ijms-22-09549-f003:**
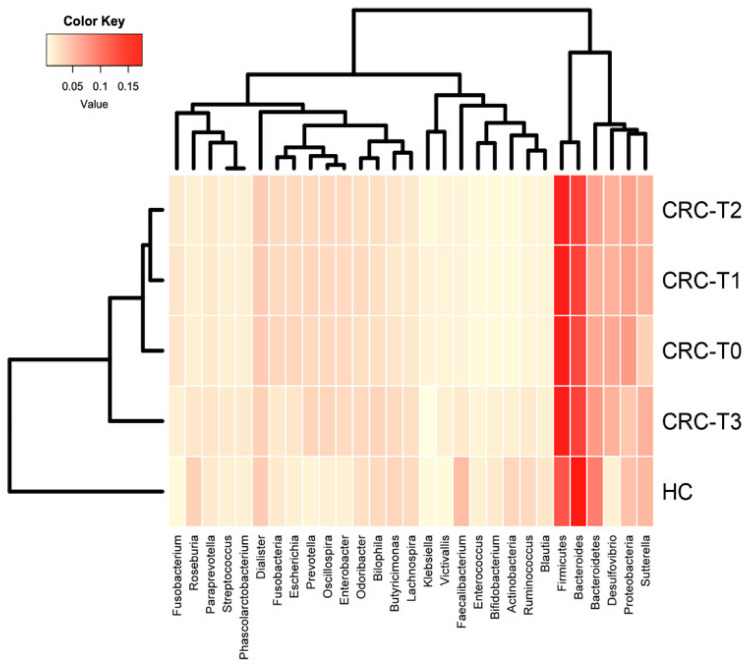
Heatmap diagram of the gut microbiota composition at different taxa levels for baseline (CRC-T0), treatment points with neoadjuvant RCT (CRC-T1, CRC-T2 and CRC-T3), and the healthy control subjects (HC). The 29 phylum and genera that were shared by all of the tested samples (core microbiome) are displayed.

**Figure 4 ijms-22-09549-f004:**
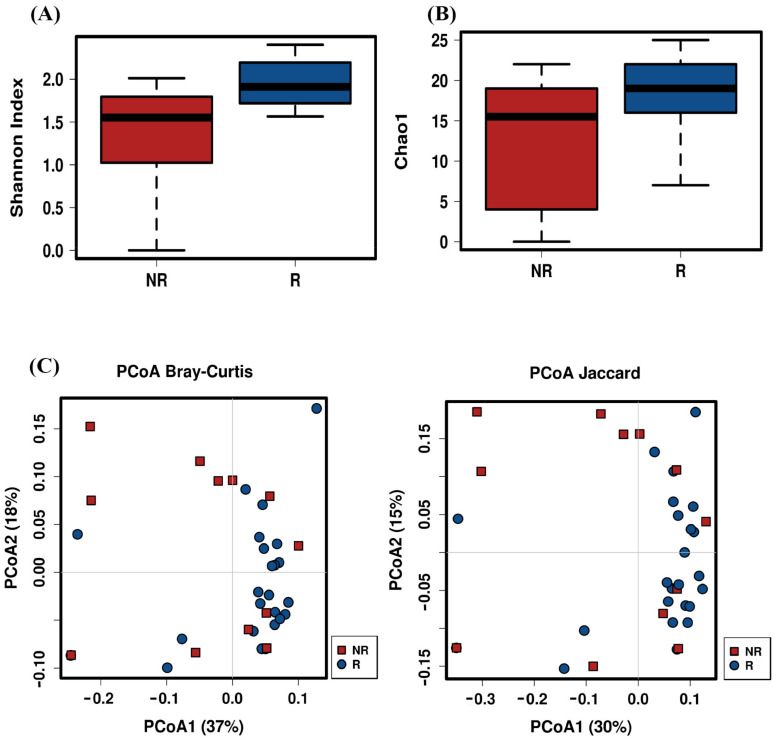
Comparison of alpha and beta diversity in CRC patients according to their response to therapy. (**A**) Shannon index; (**B**) Chao1index; (**C**) principal component plot based on the Bray–Curtis distance matrix and the Jaccard indices from the responder (R) and non-responder (NR) patients at genus-level. The first two coordinates are plotted with the percentage of variability, which is explained and indicated on the axis.

**Figure 5 ijms-22-09549-f005:**
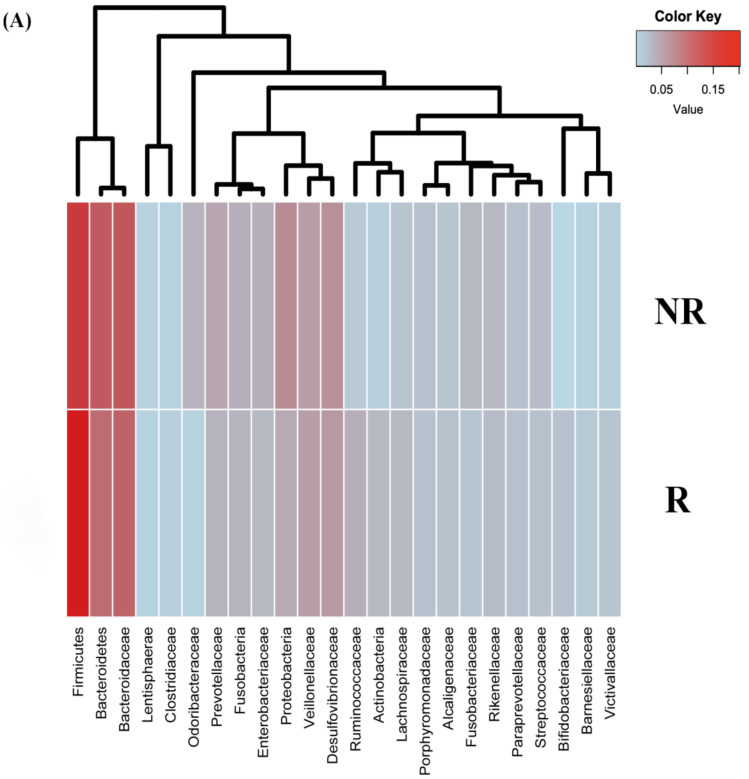
Heatmap of the fecal microbiota composition at the phylum and family levels in the responder (R) and non-responder (NR) patients (**A**). Relative abundance at phylum (**B**) and family (**C**) levels of differentially abundant OTUs in the stool samples of N patients compared to the NR patients. * *p* < 0.05.

**Figure 6 ijms-22-09549-f006:**
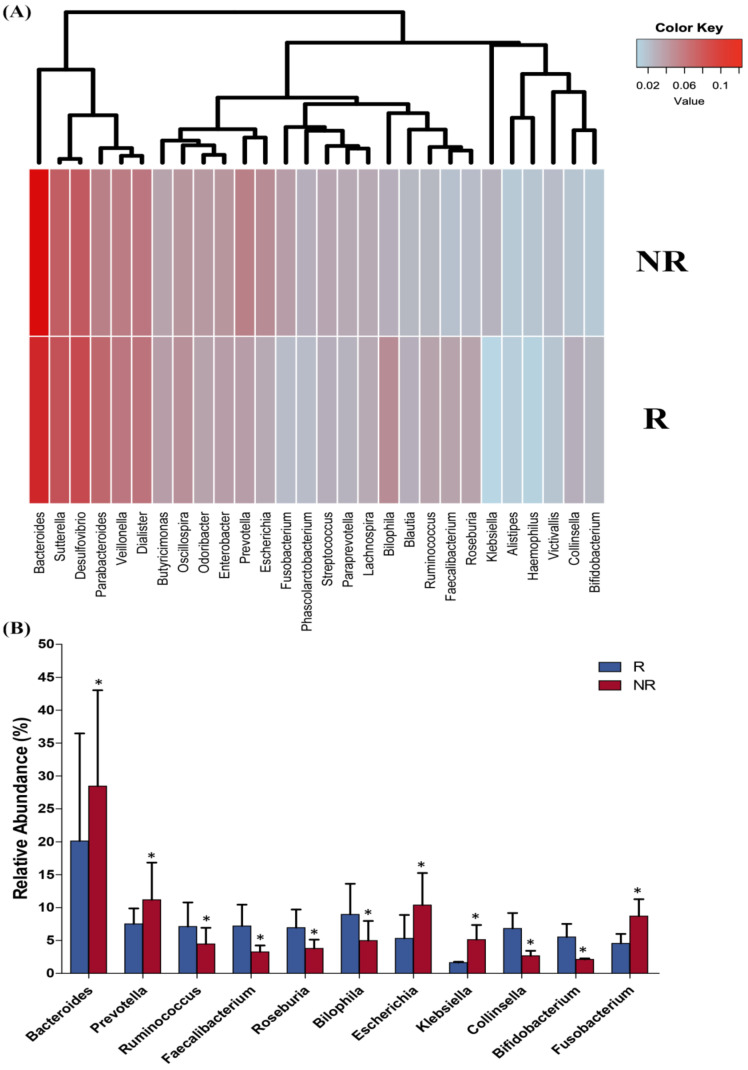
Heatmap of the fecal microbiota composition at genera level in the responder (R) and non-responder (NR) patients (**A**)**.** Relative abundance at genera level of differentially abundant OTUs in the stool samples of the N patients compared to the NR patients. * *p* < 0.05 (**B**).

**Figure 7 ijms-22-09549-f007:**
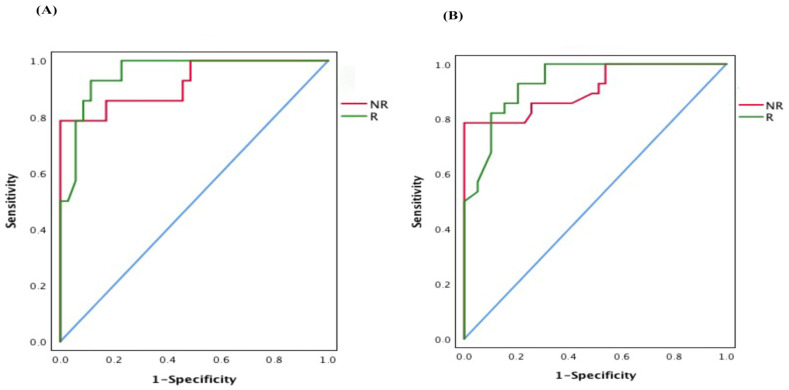
Receiver operating characteristic (ROC) curve based on the random forest classifier constructed using microbial variables (*Ruminococcus albus, Bifidobacterium bifidum*, *Faecalibacterium prausnitzii*, *Fusobacterium nucleatum*, and *Bacteroides fragilis*). (**A**) Training cohort. The area under the ROC curve (AUC) was 0.95, and the 95% confidence interval (CI) was 0.901–1 for the R patients (green), and the AUG was 0.92 and 95% the IC was 0.827–1 for the NR patients (red). (**B**) Validation cohort. The AUG was 0.93 and the 95% IC was 0.877–0.987 for the R patients (green), and the AUG was 0.91 and 95% the IC was 0.835–0.984 for the NR patients (red).

**Figure 8 ijms-22-09549-f008:**
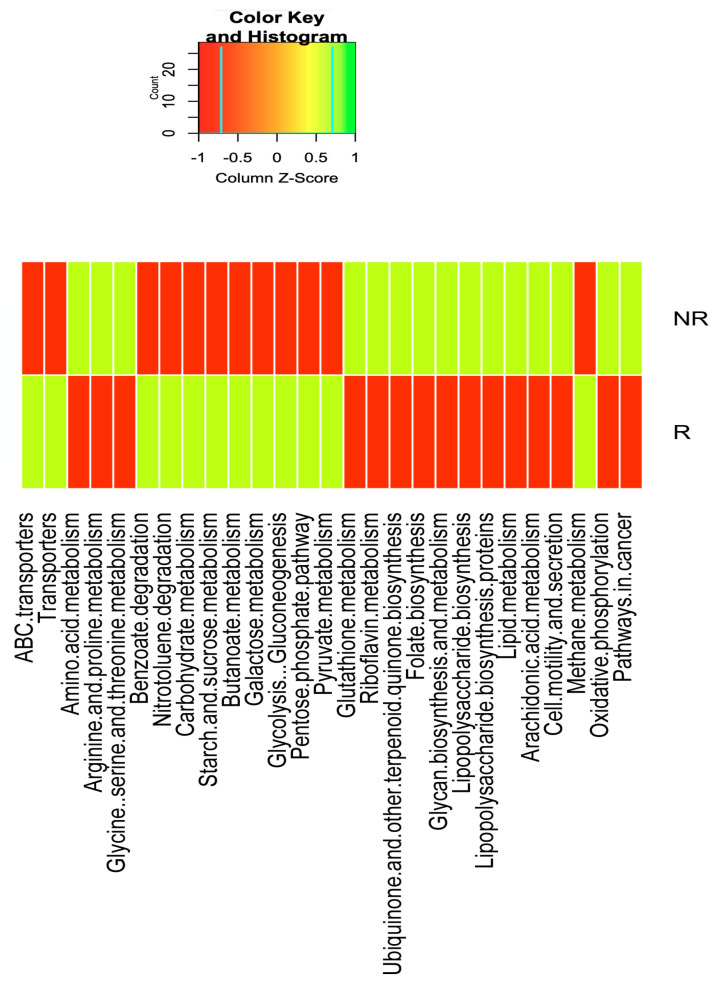
Heatmap of bacterial gene functional predictions using the PICRUSt algorithm from the fecal samples from the responder (R) patients and the non-responder (NR) patients.

**Table 1 ijms-22-09549-t001:** Clinical characteristics of study groups.

	Healthy Controls (N = 20)	CRC-Patients (N = 40)	** p*	R Patients (N = 28)	NR Patients (N = 12)	** p*
Age (years)	61.42 ± 7.40	63.35 ± 6.97	0.326	62.93 ± 8.27	63.12 ± 6.34	0.928
Gender, n (M/F)	10/10	23/17	0.783	16 /12	7/5	0.780
BMI (kg/m^2^)	25.45 ± 3.23	26.42 ± 4.71	0.412	26.22 ± 4.22	25.92 ± 3.92	0.835
Constipation, n (%)	6 (20%)	10 (25%)	0.914	7 (25%)	3 (25%)	0.690
Alcohol consumption, n (%)	4 (13.3%)	6 (15%)	0.831	4 (14.28%)	2 (16.16%)	0.740
Current smoking, n (%)	9 (30%)	15 (37.5%)	0.774	11 (39.28%)	4 (33.33%)	0.990
Biochemical data						
Glucose (mg/dl)	94.85 ± 19.86	104.79 ± 27.94	0.161	102.83 ± 26.38	104.15 ± 23.56	0.882
Total cholesterol (mg/dl)	175.2 ± 33.6	183.95 ± 25.71	0.268	184.17 ± 21.64	181.67 ± 26.12	0.755
Triglycerides (mg/dl)	112.67 ± 34.51	114.85 ± 33.62	0.815	109.25 ± 32.12	118.32 ± 27.12	0.398
HDL-cholesterol (mg/dl)	60.7 ± 15.1	54.83 ± 18.23	0.219	55.32 ± 16.21	53.89 ± 18.34	0.807
LDL-cholesterol (mg/dl)	107.78 ± 27.12	112.07 ± 33.45	0.621	109.68 ± 30.29	112.36 ± 33.21	0.805
Histological variables						
Disease stage						
II		22 (55%)	-	15 (53.57%)	7 (58.33%)	0.945
III		18 (45%)	-	13 (46.42%)	5 (41.66%)	0.950
Tumor depth penetration (T)						
T2–T3		26 (65%)	-	18(64.28%)	8 (66.66%)	0.828
T4		14 (35%)	-	10 (35.71%)	4 (33.33%)	0.832
Grade of differentiation						
G1		18 (45%)	-	12 (42.85%)	6 (50%)	0.944
G2		10 (25%)	-	7 (25%)	3 (25%)	0.690
G3		7 (17.5%)	-	5 (18.85%)	2 (16.16%)	0.806
No differentiation		5 (12.5%)	-	3 (10.71%)	2 (16.66%)	0.777

CRC: colorectal cancer; R: responders; NR: non-responders; BMI: body mass index; HDL: high density lipoprotein; LDL: low density lipoprotein. Values are expressed as mean ± SD. * *p* < 0.05 was considered statistically significant.

**Table 2 ijms-22-09549-t002:** Serum polyamines levels at baseline (T0) and post-treatment (T3).

	R Patients (N = 28)	NR Patients (N = 12)	Between-Group Difference ^1^	*p* ^2^
Agmatine (ng/mL) Baseline Post-treatment Change	0.11 ± 0.13 0.25 ± 0.24 0.14 (−0.27, −0.13)	0.13 ± 0.15 0.17 ± 0.15 0.035 (−0.13, 0.061)	0.025 (−0.11, 0.63)	0.571
Arginine (μg/mL) Baseline Post-treatment Change	23.18 ± 4.20 22.82 ± 4.16 −0.36 (−1.5, 2.27)	24.54 ± 4.76 23.10 ± 4.48 −1.43 (−1.13, 4.0)	−1.35 (−4.05, 1.35)	0.319
Ornithine (μg/mL) Baseline Post-treatment Change	19.46 ± 5.74 20.21 ± 4.16 0.74 (−3.69, 2.19)	23.31 ± 8.06 22.80 ± 7.55 −0.51 (−3.72, 4.74)	−3.85 (−8.07, 0,37)	0.073
N1,N12-diacetylspermine (ng/mL) Baseline Post-treatment Change	1.08 ± 0.43 0.90 ± 0.52 −0.18 (0.017, 0.34)	1.68 ± 1.34 1.22 ± 0.57 0.46 (−0.152, 1.07)	−0.59 (−1.20, 0.06)	0.015
N1,N8-diacetylspermidine (ng/mL) Baseline Post-treatment Change	0.71 ± 0.26 0.74 ± 0.34 0.03 (−0.13, 0.059)	0.99 ± 1.03 0.88 ± 0.38 −0.11 (−0.34, 0.57)	−0.28 (−0.74, 0.17)	0.007
N1-acetylspermidine (ng/mL) Baseline Post-treatment Change	22.47 ± 7.10 23.42 ± 8.26 0.94 (−3.88, 1.99) *	27.68 ± 13.47 28.89 ± 10.38 1.20 (−6.10, 3.68) *	−5.21 (−11.73, 1.3)	0.021
N8-acetylspermidine (ng/mL) Baseline Post-treatment Change	14.52 ± 3.48 14.69 ± 3.39 0.16 (−0.90, 0.57)	14.88 ± 3.27 16.10 ± 2.33 1.22 (−2.42, −0.20) *	−0.35 (−2.38, 1.67)	0.727
N1-acetylputrescine (ng/mL) Baseline Post-treatment Change	5.04 ± 1.60 4.77 ± 1.70 −0.27 (−1.78, 1.09)	5.92 ± 5.38 5.39 ± 3.79 −0.53 (−1.01, 3.32)	−0.88 (−3.29, 1.53)	0.030
Putrescine (ng/mL) Baseline Post-treatment Change	8.84 ± 4.40 8.06 ± 3.89 −0.78 (−0.39, 1.96)	7.95 ± 3.52 7.47 ± 3.09 −0.47 (−1.07, 2.02)	0.89 (−1.49, 3.28)	0.457
Spermidine (ng/mL) Baseline Post-treatment Change	17.14 ± 7.19 20.42 ± 12.40 3.28 (−7.42, 0.85)	22.26 ± 12.69 20.90 ± 10.81 −1.35 (−2.01, 4.73)	−4.11 (−11.36, 1.12)	0.106
N1-acetylspermine (ng/mL) Baseline Post-treatment Change	0.89 ± 0.33 1.19 ± 0.63 0.29 (−0.55, −0.046)	1.48 ± 0.70 1.33 ± 0.62 −0.14 (−0.11, 0.40)	−0.58 (−0.92, −0.25)	0.014
Spermine (ng/mL) Baseline Post-treatment Change	3.77 ± 1.30 4.80 ± 2.88 1.03 (−2.17, 0.107) *	12.10 ± 7.85 7.35 ± 3.66 −4.74 (1.71, 7.77) *	−7.32 (−11.74, −4,89)	0.001

Serum polyamine levels were measured by means of ultra-high performance liquid chromatography tandem mass spectrometry (UHPLC-MS/MS). Values are expressed as mean ± SD or mean (95% CI). R: responder; NR: non-responder. ^1^ Difference between R and NR patients at post-treatment when adjusted for baseline. ^2^ Comparison among post-treatment changes was conducted with a covariance model (ANCOVA) adjusted for baseline. * Wilcoxon signed-rank test was used to calculate differences in polyamines between baseline and post-treatment in R and NR patients. *p* < 0.05 was considered statistically significant.

**Table 3 ijms-22-09549-t003:** Fecal SCFAs concentrations and serum zonulin levels at baseline (T0) and post-treatment (T3).

	R Patients (N = 28)	NR Patients (N = 12)	Between-Group Difference ^1^	*p* ^2^
Acetic acid (mg/g) Baseline Post-treatment Change	0.83 ± 0.39 1.04 ± 0.40 0.20 (−0.39, 0.31) *	0.71 ± 0.15 0.77 ± 0.17 0.06 (−0.30, 0.18)	0.26 (−0.03, 0.56)	0.012
Propionic acid (mg/g) Baseline Post-treatment Change	1.40 ± 1.27 1.01 ± 1.10 −0.39 (−0.51, 0.59)	2.02 ± 1.35 1.70 ± 1.52 −0.32 (−0.9, 0.36)	−0.68 (−0.86, 1.76)	0.102
Butyric acid (mg/g) Baseline Post-treatment Change	1.37 ± 0.45 2.36 ± 1.82 0.99 (−1.2, 2.15) *	0.93 ± 0.68 1.02 ± 1.07 0.09 (−0.65, 1.34)	1.33 (−0.04, 2.71)	0.016
Isobutyric acid (mg/g) Baseline Post-treatment Change	0.58 ± 0.33 0.69 ± 0.05 0.11 (0.07, 0.21)	0.31 ± 0.33 0.44 ± 0.15 −0.13 (−0.23, 0.76)	0.15 (0.03, 0.26)	0.010
Valeric acid (mg/g) Baseline Post-treatment Change	0.30 ± 0.16 0.13 ± 0.07 −0.17 (−0.27, 0.39)	0.61 ± 0.32 0.29 ± 0.19 -0.47 (−0.58, 0.76)	−0.25 (−0.38, 0.29)	0.002
Isovaleric acid (mg/g) Baseline Post-treatment Change	0.50 ± 0.49 0.20 ± 0.13 −0.30 (−0.43, 0.31)	0.90 ± 0.44 0.39 ± 0.24 −0.51 (0.66, 1.02)	−0.18 (−0.45, 0.29)	0.009
4-methylvaleric acid (mg/g) Baseline Post-treatment Change	0.13 ± 0.23 0.07 ± 0.10 −0.06 (−0.09, 0.15)	0.37 ± 0.64 0.04 ± 0.01 −0.33 (−0.47, 0.86)	0.20 (−0.35, 0.10)	0.216
Hexanoic acid (mg/g) Baseline Post-treatment Change	0.15 ± 0.20 0.10 ± 0.10 −0.04 (−0.09, 0.10)	0.11 ± 0.08 0.05 ± 0.09 −0.05 (−0.07, 0.13)	0.05 (−0.19, 0.13)	0.007
Heptanoic acid (mg/g) Baseline Post-treatment Change	0.09 ± 0.15 0.06 ± 0.06 −0.03 (−0.06, 0.07)	0.07 ± 0.06 0.05 ± 0.01 −0.02 (−0.04, 0.08)	0.02 (−0.07, 0.04)	0.171
Zonulin (ng/mL) Baseline Post-treatment Change	257.6 ± 65.4 218.1 ± 76.4 −39.3 (−52.2, 23.9)	272.6 ± 35.1 298.4 ± 47.5 25.2 (11.3, 37.1)	−22.2 (−37.4, 10.2)	0.004

Short-chain fatty acids (SCFAs) in fecal samples were analyzed by means of gas chromatography coupled with a flame-ionization detector (GC-FID). Values are expressed as mean ± SD or mean (95% CI). R: responder; NR: non-responder. ^1^ Difference between R and NR patients at post-treatment when adjusted for baseline. ^2^ Comparison among post-treatment changes was conducted with a covariance model (ANCOVA) adjusted for baseline. * Wilcoxon signed-rank test was used to calculate differences in the SCFAs and zonulin between the baseline and post-treatment in R and NR patients. *p* < 0.05 was considered statistically significant.

## Data Availability

The data presented in this study are available upon request from the corresponding author. The data are not publicly available, as they contain information that could compromise the privacy of research participants.

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
