# Peer review of "Relationships of Gut Microbiota Composition, Short-Chain Fatty Acids and Polyamines with the Pathological Response to Neoadjuvant Radiochemotherapy in Colorectal Cancer Patients"

_ijms, 2021, doi:10.3390/ijms22179549_

Round 1

Reviewer 1 Report

I think the authors respond the requests properly.

Reviewer 2 Report

Not comments

Reviewer 3 Report

The present manuscript has been improved in the text and Figure organization meaning that, at least regarding my initial queries, the authors have  provided satisfactory revision.

This manuscript is a resubmission of an earlier submission. The following is a list of the peer review reports and author responses from that submission.

Round 1

Reviewer 1 Report

This paper relates to a leading topic in the field: the contribution of gut microbiota in drug response, in the context of colorectal cancer (CRC). In this study, authors focused on radiochemotherapy outcome (radiotherapy + 5-FU) by focusing on gut taxonomy, but also more interestingly in gut metabolic functions.

Language: Overall, the manuscript reads well, the EN language is good and I found no obvious typos. This is appreciated. I would not request additional reviewing on that aspect.

Ethics: Ethical aspects seem to be in place, with ethic protocols approved by their national institution (Medical Ethics Committee at the Virgen de la Victoria University Hospital).

Data availability: Overall, authors have chosen bar graphs, and tables and heatmap matrices, making it difficult to visualize individual values if any deeper investigation is required. Raw data are not publicly available in a repository database, but are available upon request. Still, I believe confidential clinical details from patients should not be an issue to remove and should not prevent public deposition of raw data. I let the editorial office to decide if they believe this comment is relevant.

Method section is precise and appropriate.

Discussion: Literature is up to date, this is appreciated. Although I believe this section is a bit long, most of it is relevant to the findings and reads well.

The authors are invited to clarify several specific points to reinforce the findings they are reporting here:

- To me it is unclear if and how authors assessed eating habits of their clinical cohort.

- Please detail the handling of the human feces for sequencing: DNA extracted from frozen or fresh material? superficial or deep sampling?

- It would be helpful for the reader to have visible statistics in corresponding panels (see Figure 1 panels A and B), alternatively, it could be added in Figure legends.

- Overall, the first part of the findings focused on microbiome taxonomy is very descriptive, and I don't feel there is a need for 6 different Figures here. A main figure with only beta diversity, and bar graph for most discriminant taxa would be enough. Differentially abundant phyla analysis Figure 2 is interesting, and in many aspects fit with current knowledge in the field, although I am troubled by the relative abundance scale. I would have expected normalized values to 1. This would make the understanding of such values easier to comprehend. LEfSe analysis could have been a nice addition here.

- Results 2.5, this part is very elusive to me, and I need a corresponding Figure.

- Results 2.6, I recognize that inferential analysis for microbiome metabolome function is interesting, and can be performed quickly with minimal knowledge (eg. 16S), but it is very speculative and not much convincing for the bacteriologist reader. At least for some "metabolites/metabolic functions", a proper biological confirmation is needed. This is also why I don't see the connection with this part and the subsequent dosage of polyamines in results 2.7.

- Results 2.7. Even if the method part is accurate, I recommend to precise in Table legends the technical approach for dosage of blood metabolites.

Author Response

This paper relates to a leading topic in the field: the contribution of gut microbiota in drug response, in the context of colorectal cancer (CRC). In this study, authors focused on radiochemotherapy outcome (radiotherapy + 5-FU) by focusing on gut taxonomy, but also more interestingly in gut metabolic functions.

Language: Overall, the manuscript reads well, the EN language is good and I found no obvious typos. This is appreciated. I would not request additional reviewing on that aspect.

Ethics: Ethical aspects seem to be in place, with ethic protocols approved by their national institution (Medical Ethics Committee at the Virgen de la Victoria University Hospital).

Data availability: Overall, authors have chosen bar graphs, and tables and heatmap matrices, making it difficult to visualize individual values if any deeper investigation is required. Raw data are not publicly available in a repository database, but are available upon request. Still, I believe confidential clinical details from patients should not be an issue to remove and should not prevent public deposition of raw data. I let the editorial office to decide if they believe this comment is relevant.

Method section is precise and appropriate.

Discussion: Literature is up to date, this is appreciated. Although I believe this section is a bit long, most of it is relevant to the findings and reads well.

Response : First of all, we would like to thank the reviewer for his/her useful comments and suggestions, which undoubtedly have helped us to improve the manuscript.

The authors are invited to clarify several specific points to reinforce the findings they are reporting here:

-  Comment: To me it is unclear if and how authors assessed eating habits of their clinical cohort.

Response: Thank you for your pertinent comment. All the CRC patients and healthy controls included in our study followed a Mediterranean dietary pattern, and the adherence to the Mediterranean diet was assessed via a validated 14-item food frequency questionnaire (Martinez-Gonzalez et al. PLoS One. 2012;7(8):e43134). This information has been added in the Results of the revised manuscript (Lines 103-107):

“CRC patients and healthy controls followed a Mediterranean diet, consisting in a high consumption of olive oil, fruits, legumes, vegetables, nuts, whole grains and fish, and a low intake of red meat and dairy products. Adherence to the Mediterranean diet was assessed by using a validated 14-item food frequency questionnaire (Martínez-González, M et al. PLoS ONE 2012;7(8):e43134) in all study patients”.

- Comment: Please detail the handling of the human feces for sequencing: DNA extracted from frozen or fresh material? superficial or deep sampling?

Response: Thank you for your comment. As recommended by the reviewer we have included information about the handling of the human feces for sequencing. This information has been added in the M&M section of the revised manuscript (Lines 568-571):

“Frozen fecal samples were thawed at 4°C to avoid dramatic temperature changes that might affect bacterial DNA integrity. Afterwards, fecal samples were manually homogenized for 30 seconds with a sterile plastic scoop and aliquots of 200 mg were used for DNA extraction using the QIAamp DNA Stool Mini kit following the manufacturer's in-structions (Qiagen, Hilden, Germany).”

- Comment: It would be helpful for the reader to have visible statistics in corresponding panels (see Figure 1 panels A and B), alternatively, it could be added in Figure legends.

Response: As is indicated by the reviewer, statistics have been added in the legends of the Figure S1A, B and C and Figure 2A, B and C.

Comment: Overall, the first part of the findings focused on microbiome taxonomy is very descriptive, and I don't feel there is a need for 6 different Figures here. A main figure with only beta diversity, and bar graph for most discriminant taxa would be enough. Differentially abundant phyla analysis Figure 2 is interesting, and in many aspects fit with current knowledge in the field, although I am troubled by the relative abundance scale. I would have expected normalized values to 1. This would make the understanding of such values easier to comprehend. LEfSe analysis could have been a nice addition here.

Response: Thanks for the suggestion. We have now reduced the number of figures in the revised manuscript by moving Figure 1 to Supplementary Material, and by merging Figures 3 and 4 (Figure 2 in the new version of the manuscript). Also, we have added a heatmap diagram of the gut microbiota composition at different taxa levels for the different treatment points (new Figure 3) and for both responders (R) and non-responders (NR) patients (new Figure 4A and 5A).  Finally, we have changed the relative abundance scales in all the figures for differentially abundant phyla, genera and families to percentages (now Figure 1, 5B and C and 6B).

- Comment: Results 2.5, this part is very elusive to me, and I need a corresponding Figure.

Response:  The corresponding figure to results 2.5 (new Figure 7) has been added in the revised manuscript.

- Comment: Results 2.6, I recognize that inferential analysis for microbiome metabolome function is interesting, and can be performed quickly with minimal knowledge (eg. 16S), but it is very speculative and not much convincing for the bacteriologist reader. At least for some "metabolites/metabolic functions", a proper biological confirmation is needed. This is also why I don't see the connection with this part and the subsequent dosage of polyamines in results 2.7.

Response: In our study we have used the PICRUSt software to estimate the metabolic capacity of the microbiome contained in 16S libraries. This software takes into account several factors which are important for metagenomic prediction, such as the availability of pan and core genomes of microbial reference taxa (Collins RE, et al. Mol Biol Evol 2012, 29:3413-25) and 16S rRNA copy number among bacterial taxa (VÄ›trovský T, et al. PLoS ONE 2013, 8:e57923). In our PICRUST analysis we found a significant over-representation of genes for amino acid metabolism pathways (including arginine metabolism) in NR compared to R patients. A recent metagenomic analysis has established that the CRC-associated microbiome showed an association with the conversion of arginine to polyamines, gut microbiota derived metabolites with pro-carcinogenic effects (Thomas et al. Nat. Med. 2019;25:667–678). This is particularly important in the case of NR patients, which is surrounded by intestinal bacteria that are able to produce high levels of polyamines such as Bacteroides fragilis and Fusobacterium nucleatum. Then, PICRUSt could provide insights into what additional putative microbial pathway changes might explain or accompany the compositional variation between R and NR patients in our study. This information has been added in the Discussion of the revised manuscript (Lines 456-458):

“Remarkably, a recent metagenomic analysis reported that the CRC-associated microbiome showed an association with the conversion of amino acids into polyamines (e.g., the biosynthesis of putrescine from the amino acids L-arginine and L-ornithine), indicating that these metabolites could be particularly important in CRC development and progression”

- Comment: Results 2.7. Even if the method part is accurate, I recommend to precise in Table legends the technical approach for dosage of blood metabolites.

- Response: As suggested by the reviewer, we have now described the technical approach used to measure the metabolites in either serum (polyamines) and stool (short-chain fatty acids) in the legends of Tables 2 and 3 (Lines 346-347 and 357-358).

Reviewer 2 Report

Dear authors the work is not

in good way.. 

The experimental part of fatty acid is not carefully described. They mentioned only short fatty acids.And the rest? So the authors have  To provide also this aspect describing  the tipology of lipids and all kind of fatty acids present in their analytes. They have To describe all the  conditions of the gc instruments ; they have to show the gc chromatograms with the identification of all tipology of  fatty acids

Author Response

Response: First of all, we thank the reviewer for his/her comments and for offering us a constructive review of our manuscript.

Comment: The experimental part of fatty acid is not carefully described. They mentioned only short fatty acids.And the rest? So the authors have To provide also this aspect describing the tipology of lipids and all kind of fatty acids present in their analytes. They have To describe all the conditions of the gc instruments; they have to show the gc chromatograms with the identification of all tipology of fatty acids.

Response: In this study we have only measured several short chain fatty acids (SCFAs) in stool, as they are tumor-suppressing metabolites specifically produced by colonic microbiota. Moreover, previous studies have shown that fecal SCFA levels were lower in CRC patients compared to control, which was associated to a decrease in the number of SCFAs-producing bacteria in these patients. Therefore, in our research we aimed to study only the association between the stool levels of the main short-chain fatty acids (SCFAs) with the pathological complete response to neoadjuvant radiochemotherapy in CRC patients.

In addition, we have now added in the M&M section the conditions of the GC instrument used for SCFA determination (Lines 602-622). Unfortunately, as we measured the SCFAs through an external service, the GC chromatograms are not currently available, at least until September. However, whether the editor considers that this is crucial for publication we will contact to the external service to get this information as soon as possible .

Finally, the analysis of serum fatty acids is beyond the scope of this work, as they are not directly produced by the gut microbiota. Nevertheless, we appreciate the reviewer’s suggestion and we will consider to study whether the profile of serum fatty acids is related to the response to neoadjuvant radiochemotherapy in CRC patients.

Reviewer 3 Report

This study aimed to evaluate the potential contribution of the gut microbiota and microbial-derived metabolites such as short chain fatty acids and polyamines to neoadjuvant radiochemotherapy (RCT) outcome in CRC patients. The authors showed that the fecal microbiota of responders was enriched in butyrate producing-bacteria and have significantly higher level of acetic, butyric, isobutyric, and hexanoic acids than non-responders. Some major concerns are listed as followings:

  1. The rationale to test neoadjuvant radiochemotherapy (RCT) outcome in CRC patients is not so strong.
  2. Overall sample size (only 40 cases of CRC) for the current study is low, thus the results might not be reliable given inadequate power of the study. An a priori power calculation would have been useful.
  3. For the selection and identification of patients with CRC for this study, more information on the Inclusion/Exclusion criteria needs to be included.
  4. The limitation should be mentioned in the discussion section.
  5. Scientific writing and careful English grammar editing would also improve this paper.

Author Response

This study aimed to evaluate the potential contribution of the gut microbiota and microbial-derived metabolites such as short chain fatty acids and polyamines to neoadjuvant radiochemotherapy (RCT) outcome in CRC patients. The authors showed that the fecal microbiota of responders was enriched in butyrate producing-bacteria and have significantly higher level of acetic, butyric, isobutyric, and hexanoic acids than non-responders.

Response: First of all, we thank the reviewer for his/her comments and for offering us a constructive review of our manuscript.

Some major concerns are listed as followings:

Comment: The rationale to test neoadjuvant radiochemotherapy (RCT) outcome in CRC patients is not so strong.

Response: Thanks to the reviewer for his/her pertinent comment. We have now highlighted that the rationale of this study by adding the following text: (Lines 91-95).

“Therefore, taking all the evidence together we hypothesized a bidirectional interaction between the neoadjuvant RCT and the gut microbiome in CRC patients: RCT might induce alterations in the gut microbiome; and that these alterations might in turn influence the effectiveness of RCT by directly interact with the treatment and/or by stimulating the host’s immune response”. 

Comment: Overall sample size (only 40 cases of CRC) for the current study is low, thus the results might not be reliable given inadequate power of the study. An a priori power calculation would have been useful.

Response: We totally agree with the reviewer that the sample size used in the present study is small. Nevertheless, the sample size of 40 cases of CRC patients was not random, since we calculated a priori the necessary sample size to achieve sufficient statistical power, based on a previous paper from Yang et al. Theranostics 2019; 9(14): 4101-4114, were changes in the intestinal microbiota and metabolome (i.e. polyamines) between CRC patients and healthy subjects were evaluated. In this study, the authors found a difference in the proportion of Proteobacteria of 16.7%. With these data, and assuming an alpha error of 0.05, a power of 80% and a loss rate of 15% (loss of samples or other reasons not related to the intervention), a total of 23 patients and controls would need to be included.  Also, we have based our sample size in recent studies with lower or similar number of cases an controls that have analyzed the predictive value of gut microbiome in terms of response after preoperative chemoradiotherapy (Bum-Sup et al. Int J Radiation Oncol Biol Phys, 2020;07(4):736-746) with sample size of 45 CRC patients, Cong et al. Frontiers in Microbiology 2019;10:1008 with a sample size of 22 CRC patients and 21 healthy controls or  Cong et al. Frontiers in Microbiology 2018;9:2777 with a  sample size of 10 CRC patients and 11 healthy controls). 

Comment: For the selection and identification of patients with CRC for this study, more information on the Inclusion/Exclusion criteria needs to be included.

Response: As is suggested by the reviewer we have included more information on the inclusion and exclusion criteria of the CRC patients (Lines 535-543).

Comment: The limitation should be mentioned in the discussion section.

 Response: We thank the reviewer for his/her comment. We have included the study limitations and strengths in the Discussion section of the revised manuscript (Lines 522-530).

“This study has some limitations such as a relatively small sample size, which could reduce the power of the study. However, despite the relatively small size of our study, statistically significant differences were observed, suggesting that the results presented herein provide solid evidence on the potential contribution of the gut microbiome in the RCT outcomes in CRC patients. Besides, our study also has some several strengths such as the careful design, the well-matched cohorts of CRC patients and controls, a complete definition of inclusion and exclusion criteria, and the consideration of lifestyle-associated confounders factors that may affect the gut microbiota composition such as the dietary pattern”.

Comment: Scientific writing and careful English grammar editing would also improve this paper.

Response: The paper has been edited and English grammar has been revised.

Round 2

Reviewer 2 Report

Dear Authors

based on your replay  and on the new version of the experimental part I got that you  have serious flaws about the analysis and characterizazion of fatty acids and short chain fatty acids.

Reviewer 3 Report

I think the authors respond the requests properly.